# Transformation of Tetracycline by Manganese Peroxidase from *Phanerochaete chrysosporium*

**DOI:** 10.3390/molecules26226803

**Published:** 2021-11-11

**Authors:** Xuemei Sun, Yifei Leng, Duanji Wan, Fengyi Chang, Yu Huang, Zhu Li, Wen Xiong, Jun Wang

**Affiliations:** 1School of Civil Engineering, Architecture and Environment, Hubei University of Technology, Wuhan 430068, China; sunxuemei@hbut.edu.cn (X.S.); duanjiwan@hbut.edu.cn (D.W.); chang602118@163.com (F.C.); huangyu@hbut.edu.cn (Y.H.); iamlizhu@163.com (Z.L.); wenx2217@163.com (W.X.); 2Hubei Key Laboratory of Ecological Restoration for River-Lakes and Algal Utilization, Wuhan 430068, China; 3College of Marine Sciences, South China Agricultural University, Guangzhou 510642, China; wangjun2016@scau.edu.cn; 4Institute of Eco-Environmental Research, Guangxi Key Laboratory of Marine Natural Products and Combinatorial Biosynthesis Chemistry, Biophysical and Environmental Science Research Center, Guangxi Academy of Sciences, Nanning 530007, China

**Keywords:** tetracycline, manganese peroxidase, antibacterial potency, transformation products, transformation mechanism

## Abstract

The negative impacts on the ecosystem of antibiotic residues in the environment have become a global concern. However, little is known about the transformation mechanism of antibiotics by manganese peroxidase (MnP) from microorganisms. This work investigated the transformation characteristics, the antibacterial activity of byproducts, and the degradation mechanism of tetracycline (TC) by purified MnP from *Phanerochaete chrysosporium*. The results show that nitrogen-limited and high level of Mn^2+^ medium could obtain favorable MnP activity and inhibit the expression of lignin peroxidase by *Phanerochaete chrysosporium*. The purified MnP could transform 80% tetracycline in 3 h, and the threshold of reaction activator (H_2_O_2_) was about 0.045 mmol L^−1^. After the 3rd cyclic run, the transformation rate was almost identical at the low initial concentration of TC (77.05–88.47%), while it decreased when the initial concentration was higher (49.36–60.00%). The antimicrobial potency of the TC transformation products by MnP decreased throughout reaction time. We identified seven possible degradation products and then proposed a potential TC transformation pathway, which included demethylation, oxidation of the dimethyl amino, decarbonylation, hydroxylation, and oxidative dehydrogenation. These findings provide a novel comprehension of the role of MnP on the fate of antibiotics in nature and may develop a potential technology for tetracycline removal.

## 1. Introduction

Tetracycline (TC) antibiotics are widely used in human health and veterinary medicine because of their low price and solid antibacterial ability [1]. China is one of the largest producers and consumers of antibiotics globally [2], and the top four frequently used antibiotics are tetracyclines [3]. Most antibiotics taken by humans or animals are released into the environment in their original drug forms and metabolites due to their low absorption rate [4]. According to Qian-Qian et al., the total amount of antibiotics emissions reached 53,800 tons in China 2013 [5]. Recently, residual tetracyclines in various environments have been reported frequently, including surface water [6], marine [7], sediments [8], soil, and even in biota samples [9]. Tetracyclines have detrimental effects on ecosystems, such as bioaccumulation along the food chain, toxicity to the microbial community, promotion of the generation and dissemination of antibiotic resistance (especially in pathogens) [10], threat to drinking sources and irrigation water, and disruption of the human intestinal flora. These negative impacts may pose a potential threat to human health, causing a big concern regarding tetracycline contaminants.

Previous studies have demonstrated that hydrolysis, photolysis, ozonation, chlorination, and advanced oxidation could effectively remove tetracycline. However, these technologies not only consume lots of energy or reagents but may also generate hazardous transformation byproducts (photolysis produced more toxic intermediate) [11]. Enzyme treatment has the advantages of mild reaction conditions, short reaction time, high efficiency, and low energy consumption, which makes it a widely used treatment of special pollutants [12]. Park et al. reported that glutathione s-transferases could degrade 30% of tetracycline in 30 min [13]. Horseradish peroxidase (from a plant) has also demonstrated the ability to degrade tetracycline, about 50% removal efficiency within 1 h [3]. *Phanerochaete chrysosporium* (one kind of white-rot fungi) is well known for being used in the removal of xenobiotic organic contaminates [14] due to the expression of three principal ligninolytic enzymes (manganese peroxidase (MnP), lignin peroxidase (LiP), and laccase) [15].

Unfortunately, most studies focused on the treatment of dyes by ligninolytic enzymes [16], and only a few reports demonstrated its capability of in vitro degradation of tetracycline. Previous studies have shown that tetracycline could be effectively removed by crude manganese peroxidase (MnP) and lignin peroxidase (LiP) expressed by white-rot fungi [17,18]. Approximately 16% tetracycline could be removed after treatment with laccase in 4 h [19]. MnP has been proven to be able to remove tetracycline. However, little is known about how the MnP system works in the transformation progress of tetracycline. Moreover, although the use of crude enzymes has become a trend for practical application of enzyme catalysis due to the high cost of the enzyme purification process [20], white-rot fungi usually produce multiple ligninolytic enzymes simultaneously [21], which may participate more or less in the degradation of tetracycline. Therefore, it is more appropriate to purify the enzyme in investigating the process and mechanism of enzymatic.

Hence, this study aims to optimize conditions of MnP production and obtain purified MnP. The transformation characteristics of tetracycline by MnP system were then investigated, including threshold and supplement of activated (hydrogen peroxide), cycle run of the MnP system, and enzymatic kinetics fitting. Finally, antimicrobial potencies of byproducts were evaluated, byproducts captured and identified, and possible degradation pathway of tetracycline by MnP system was proposed. The findings of this work provide a novel comprehension of the role of MnP on the fate of antibiotics in nature and may develop a potential technology for tetracycline removal.

## 2. Results and Discussion

### 2.1. Production and Purification of MnP

Components in media that affect *P. chrysosporium* generation of MnP were optimized by an L_8_ (2^3^) orthogonal experiment design. In different experimental groups, the times of enzyme expression were different, and the peak appeared at 7 days (run 6) (Appendix A). The enzyme activity of the 8 days was selected for orthogonal design analysis. Table 1 shows the results of variance analysis. Based on the ranges of results, the order of impact on enzymatic activity followed ammonium tartrate > glucose > MnSO_4_. The optimal concentrations were 0.2 g·L^−1^, 10 g·L^−1^, and 1 mmol·L^−1^, respectively. Under this optimal medium, the MnP activity peaks in the extracellular crude enzyme solution reached 501.17 U L^−1^. Moreover, the expression of MnP may be accompanied by the production of lignin peroxidase (LiP). However, LiP was not produced in the optimized medium (Appendix A). Therefore, the results of orthogonal design were suitable for MnP production. Higher MnP enzyme activity was obtained, and there was no LiP expression, which made the subsequent purification of MnP easier.

The variance analysis showed that the concentration of ammonium tartrate primarily influenced enzyme production (*p* < 0.05) and was negatively correlated with enzyme activity (Table 1). Previous studies have shown that MnP production is regulated by nitrogen. Some white-rot fungi, such as *Bjerkandera adusta and P. chrysosporium*, produce MnP better under nitrogen-limited conditions [22]. Relatively little nitrogen is found in the natural environment where white-rot fungi grow (wood), which leads to high levels of ligninolytic enzymes that cause wood to rot [23]. Glucose had no significant effect on MnP production. *P. chrysosporium* had the highest MnP activity under conditions of low nitrogen–high carbon, which was consistent with *Trametes* species [24]. This indicated that the ratio of carbon/nitrogen may synergistically affect the expression of MnP. Mn^2+^ is also essential for the synthesis of MnP by *P. chrysosporium* [24]. Mn^2+^ regulates MnP production by inducing gene transcription, thereby increasing levels of MnP mRNA and promoting MnP protein expression in *P. chrysosporium* [25]. Mn^2+^ had no significant effect on MnP production in this study, which may be due to the low level of Mn^2+^ concentration being enough to promote enzyme production. In addition, Mn^2+^ also had a substantial effect on the expression of LiP by *P. chrysosporium*. A low concentration of Mn^2+^ could synthesize LiP, while a high concentration repressed production of LiP [26]. This may be the reason that only MnP and no LiP was produced in the optimized medium. 

### 2.2. Degradation of TC by MnP

Figure 1a shows the degradation effect of tetracycline by MnP and the consumption of H_2_O_2_ in the process. Tetracycline degraded rapidly in 30 min, up to 66.3%. The degradation rate was then delayed and finally reached 80.9% at 3 h. The trend of H_2_O_2_ consumption and tetracycline degradation was synchronous during the reaction progress. The consumption of H_2_O_2_ slowed down after 30 min. It decreased to 0.046 mmol L^−1^ at 2 h and then stabilized. Previous studies have shown that H_2_O_2_ as an oxidant can also remove TC, but it needs a large amount of hydrogen peroxide (>29 mmol L^−1^) and an alkaline environment [27]. However, the degradation of tetracycline was not observed when only H_2_O_2_ existed (Appendix A) due to the low concentration and short reaction time, consistent with previous studies [3]. This may be due to the low concentration of H_2_O_2_ in the system, acidic pH, and short reaction time [4].

MnP is a heme-containing protein. It requires H_2_O_2_ and manganese as an activator and mediator, respectively. The catalytic cycle of MnP is illustrated in Appendix A. The native ferric MnP is oxidized to compound I via two electron steps by H_2_O_2_. Adding 1 equivalent of Mn^2+^ reduces compound I (MnP-I) to compound II (MnP-II). A second equivalent of Mn^2+^ reduces compound II (MnP-II) to native ferric MnP with the oxidation of Mn^2+^ to Mn^3+^, which mediates organic substrates’ oxidation [28]. Moreover, excess H_2_O_2_ may result in the oxidation of compound II to compound III, which is an inactivated form [18]. Therefore, H_2_O_2_ as an activator plays a critical role in the catalytic cycle. Considering the enormous consumption of H_2_O_2_ in the process of tetracycline degradation, and the structure of MnP may be disrupted when high-level of H_2_O_2_ was added at one time, H_2_O_2_ should be supplemented after a while. When the tetracycline removal trend slows down (30 min), adding H_2_O_2_ in the system could improve the degradation rate of tetracycline. The ultimate degradation rate of tetracycline increased from 83.3 to 94.7% after the H_2_O_2_ supplementation (Figure 1b).

In addition, the activity of MnP in the system changed little when the enzymatic degradation of TC reached equilibrium (Appendix A), which indicated that MnP might be reused. The residual of TC and H_2_O_2_ are shown in Figure 2 after the first catalysis and supplementation of the raw material. After adding TC and H_2_O_2_ for 10 min, H_2_O_2_ concentration decreased from 0.22 to 0.15 mmol L^−1^, TC decreased from 51.19 to 20.90 mg L^−1^, and the degradation rate was 59%. After 3 h, the concentration of H_2_O_2_ and TC decreased to 0.045 mmol L^−1^ and 13.66 mg L^−1^, respectively, and the degradation rate reached 73%. The concentration of H_2_O_2_ regulated the degradation of TC by MnP, and its threshold was about 0.045 mmol L^−1^. When the level of H_2_O_2_ was less than this threshold, the degradation of TC tended towards equilibrium. 

The degradation of TC with different concentrations by the MnP system and their cyclic degradation performance were investigated for three continuous cyclic runs (Figure 3). The results showed that the degradation rate was almost identical at low concentrations of 10 mg L^−1^ and 25 mg L^−1^ in the 3rd cyclic run and were maintained at about 77.05% and 88.47%, respectively. However, the degradation rate significantly decreased with the increase of the catalytic cycles when the initial concentration of TC was higher, e.g., the degradation rate decreased from 82.03 to 60.00% when the initial tetracycline was 50 mg L^−1^. In contrast, when the initial tetracycline increased to 100 mg L^−1^, the degradation rate decreased from 81.26 to 49.36% from the 1st to the 3rd cyclic run. This decrease may be explained by the accumulation of more degradation products of TC, which disturbed the collision between Mn^3+^ and tetracycline in a high initial concentration reaction system [4,29].

The transformation kinetics of TC by MnP can satisfactorily be described by the Michaelis–Menten model (Appendix A). The kinetic constants, *v_max_* and *K_m_*, were 20.35 mg L^−1^ min^−1^ and 238.23 mg L^−1^, respectively, based on non-linear regression analysis (Appendix A).

### 2.3. Antibacterial Potency of Tetracycline Degradation Products

The antimicrobial potency of the degradation byproducts by the MnP system at different initial concentrations of TC decreased throughout the reaction time (Figure 4). The inhibition zone diameter dropped fastest (from 18.6 to 8.2 mm) when the initial concentration of TC was 25 mg L^−1^, and the slowest when the initial concentration of tetracycline was 100 mg L^−1^. When the degradation system was without MnP at different concentrations of TC, the diameters of the inhibition zones were almost the same. This matched the result of the unsuccessful degradation of TC by H_2_O_2_ in the given reaction time (Appendix A). The toxicity and impact of pollutant degradation products are significant in their removal process. Physicochemical treatment of TC usually obtains high toxic byproducts (e.g., photolysis and electro-catalytic) [30,31]. In contrast, the byproducts of antibiotics from biotic processes such as bacteria, fungi, or enzyme catalysis had less toxicity than the original compounds [1,3,4,32]. In this work, the antibacterial activities of the MnP degradation tetracycline products were also lower than that of the parent compound (Figure 4). Hence, enzyme-catalyzed antibiotics may be more suitable than physicochemical methods because enzyme can reduce the antibacterial activity of antibiotics.

### 2.4. Transformation Products and a Possible Pathway

Tetracycline transformation products in the MnP system were captured and identified by high-resolution LC/MS/MS. The total ion chromatograms of TC degradation reaction at 0 min, 10 min, and 30 min are presented in Appendix A. Based on the difference of total ion chromatograms between the treatment and control (Appendix A), seven possible transformation products (TPs) were captured during the tetracycline degradation by MnP with *m*/*z* of 431, 417, 461, 459, 477, and 475, respectively (Table 2). Target MS/MS further analyzed all the proposed degradation products to confirm the correct structure (Appendix A). The two major fragments of TC (*m*/*z* 445) were 17 and 35 *m*/*z*, which indicated that the TC molecule generated daughter ions [M+H−NH_3_]^+^ and [M+H−NH_3_−H_2_O]^+^ by losing an NH_3_ and an H_2_O. All the proposed products have the same daughter ions via the loss of an NH_3_ and an H_2_O, confirming that the structures were correct.

The parent compound was eluted at 4.72 min with *m*/*z* of 445.1602 ([TC+H]^+^). At 0 min, a peak with the same *m*/*z* as the parent compound was identified as an epimer or isomer of tetracycline (Appendix A). 

The mass difference between TP431 and TC was 14 units produced via demethylation of the N-(CH_3_)_2_ on the C4 tertiary amine site. The N-C bond was easy to be attacked due to its low energy [27]. As shown in Appendix A, the oxidation of the dimethyl amino group by Mn^3+^ led to generation of a methylene radical, which produced a peroxyl radical via further oxidation, and formed an iminium cation. Due to its instability in an aqueous solution, the imine might undergo rapid hydrolysis to yield N-demethylation product and formaldehyde [33]. Demethylation of dimethylammonium group at C4 is a common occurrence in tetracycline treatment, such as bacteria [1], horseradish peroxidase [3], manganese oxide [34], chloramine [35], hydrogen peroxide, or hydroxyl radicals [36].

TP417 may be generated through the decarbonylation at the C1 compared to the parent TC. The process of decarbonylation is shown in Appendix A. The TC first underwent α-cleavage at the C12a-C1 bond and generated a diradical compound. This diradical compound forms another diradical intermediate by loss of CO. The intermediary could finally create the decarbonylation product through the closed ring reaction [36]. This reaction also occurred in removal of TC by peroxiredoxin in *Stenotrophomonas maltophilia* [1,37], horseradish peroxidase [3], and UV photolysis [36]. 

TP461a and TP461b may have been formed by hydroxylation of tetracycline and its iso-derivative. TP461b may be further hydroxylated to generate TP477. The C10, C11, or C12 in the phenolic-diketone and C1 or C3 in the tricarbonyl amide group of tetracycline are usually oxidized, and C10 is highly susceptible to oxidation [36,38]. Appendix A, shows that the phenolic hydroxyl group on the D ring of TC is oxidized to a phenoxy radical, yielding hydroquinone via further oxidation (*m*/*z* gain of 16). The hydroxylation may occur at C9 or C7 to form ortho or para hydroquinone. Previous studies have shown that hydroxyl group addition is most likely to appear at the neighboring C9–C10 [39]. Thus, it is most likely to generate ortho-hydroquinone group in the D ring of TC. This proposed product agreed with previous studies, which investigated the removal of TCs by MnO_2_ [38,39]. The C1 to C3 sites are other potential sites for hydroxylation [40]. It is shown in Appendix A, that the C3-enol is oxidized to a radical and then generated to a stable C3-keto [38]. Further oxidation causes the opening of the C2–C3 double bond and leaves behind a hydroxyl group at the C2 site [34]. The hydroxylation of C1–C3 ketone/enol also occurred in the removal of TC by manganese dioxide modified biochar [34], γ-Fe_2_O_3_/CeO_2_/persulfate [41], 3D polyaniline/perylene diimide via visible light irradiation [42], and superoxide anions [31].

The *ortho*-hydroquinone groups of TP461b and TP477 were further oxidative dehydrogenated to form quinone moieties TP459 and TP475, respectively (Appendix A), which were commonly observed degradations of tetracycline by bacteria [32], laccase [43], and photolysis [44]. TP459 can also generate TP477 by hydroxylation at C2. Based on the functional group differences between the above products, the possible transformation pathway of tetracycline by MnP was proposed, as illustrated in Figure 5.

## 3. Materials and Methods

### 3.1. Chemicals and Microorganism

Analytical grade tetracycline was obtained from Aladdin (Shanghai, China). Acetonitrile, methanol, and water with 0.1% formic acid (HPLC grade) were purchased from Fisher Scientifics (Pittsburgh, PA, USA). Catalase (lyophilized powder, 20,000 U mg^−1^, EC 1.11.1.6, CAS 9001-05-2) was obtained from Sigma-Aldrich (St. Louis, MO, USA). *Phanerochaete chrysosporium* (BKM-F-1767, Lyophilized powder) was purchased from China Center for Type Culture Collection (CCTCC, Wuhan, China).

### 3.2. Activation and Enzyme Production of Fungal Strain

*P. chrysosporium* lyophilized powder was activated in potato dextrose liquid medium at 37 °C until the mycelium appeared. The mycelial pellets were transferred into potato dextrose agar medium and cultured at 37 °C for one week. Pure water was then added, and it was washed and filtered by sterilized gauze to obtain spore suspension. The spore solution (1 × 10^5^ spores mL^−1^) was inoculated into an Erlenmeyer flask (250 mL) containing 100 mL enzyme production medium with sterile glass beads. The flask was cultured on a 120-rpm shaker at 37 °C. The culture was harvested under optimal conditions and centrifuged at 10,000 rpm for 10 min at 4 °C. All crude enzyme solutions were combined and stored at 4 °C until further purification. All the details of the medium are provided in the Appendix A.

### 3.3. Optimization of Enzyme Production Medium for MnP Production

The carbon source, nitrogen source, and Mn^2+^ ion concentration are the essential factors affecting the production of MnP by *P. chrysosporium*. An L_8_ (2^3^) orthogonal experiment was designed to optimize glucose, ammonium tartrate, and MnSO_4_ for MnP production using Minitab 19 software. The three components and corresponding levels were glucose (2 and 10 g L^−1^), ammonium tartrate (0.2 and 1.62 g L^−1^), and MnSO_4_ (0.02 and 1 mmol L^−1^) (Appendix A). The orthogonal array is shown in Appendix A, and a total of eight arrangements were performed. The experiments were performed in triplicate according to the above, and the MnP activity was assayed each day.

### 3.4. Purification of MnP

MnP was purified according to the previous methods [45,46]. Briefly, the crude enzyme solution was collected and filtered with filter paper, and concentrated by centrifugal ultrafiltration (30 kD Amicon Ultra-15, Millipore, Burlington, MA, USA). The concentrate was separated by DEAE-Sepharose column (Whatman DE52, Maidstone, UK) and linearly eluted with sodium acetate buffer (10 mmol L^−1^, pH 5.5) containing 0–0.5 M NaCl at 1 mL min^−1^. MnP components were merged and concentrated by centrifugal ultrafiltration, and then loaded onto a Sephadex G-75 column (Fluka, Waltham, MA, USA). The column was eluted at 0.1 mL min^−1^ with sodium acetate buffer (10 mmol L^−1^, pH 5.5). The activity components were concentrated by centrifugal ultrafiltration. The purified MnP protein exhibited a single band on SDS-PAGE (Appendix A). The purified MnP solution was stored in a fridge (4 °C) before being used. 

### 3.5. Enzyme Activity Analysis

MnP activity was assayed by spectrophotometry according to the method described by Paszczynski et al. [47]. Furthermore, 0.5 mL tartaric acid sodium tartrate buffer (0.05 mol L^−1^, pH 4.5), 0.2 mL MnSO_4_ (1 mmol L^−1^), and 0.1 mL H_2_O_2_ (1 mmol L^−1^) were mixed and placed in a 37 °C water bath for 3 to 4 min. Furthermore, 0.1 mL sample (enzyme liquid) was added, and the absorbance increase at 238 nm within 3 min was recorded. The inactivation enzyme solution was used as blank control. Oxidation of 1 μmol min^−1^ Mn^2+^ to Mn^3+^ represents one enzyme activity unit (U). The calculation equation of MnP activity is given in Appendix A.

### 3.6. Tetracycline Removal by MnP

The enzymatic degradation system of tetracycline consisted of TC (50 mg L^−1^), MnSO_4_ (0.1 mmol·L^−1^), MnP (40 U L^−1^), and H_2_O_2_ (0.2 mmol·L^−1^) in tartaric acid sodium tartrate buffer (0.05 mol·L^−1^, pH 4.5). The removal reaction was performed in a 150 mL amber glass bottle with a total volume of 50 mL on a 120-rpm shaker at 37 °C. The reaction was activated by adding H_2_O_2_ and mixing was stopped with equal volume catalase solution at different sampling times. All reactions were performed in triplicates.

### 3.7. Chemical Analysis

Agilent 1220 HPLC (Santa Clara, CA, USA) with a UV detector (355 nm) was used to detect residual tetracycline. Furthermore, 2 mL samples were passed through syringe filters (0.22 μm PTFE, Anpel, Shanghai, China). Furthermore, a 5 µL sample was injected and separated by an Agilent reversed-phase C18 column (4.6 × 150 mm i.d., 5 μm d.p.) at 35 °C in an oven. The mobile phase consisted of 22% (*v/v*) acetonitrile, 11% (*v/v*) methanol, and 67% (*v/v*) water with 0.1% formic acid. The flow rate was set at 1 mL min^−1^ with isocratic elution. 

The reaction samples were enriched and purified by Oasis HLB cartridge (6 cc/150 mg, Waters) according to the previous method [1]. Furthermore, a 5 µL sample was separated by an Agilent C18 column (4.6 × 150 mm i.d., 5 µm d.p.) and eluted equivalently at 0.3 mL min^−1^, then, the byproducts were identified by LC/MS/MS (Q Exactive Hybrid Quadrupole-Orbitrap, Thermo Scientific, Waltham, MA, USA). The Xcalibur 2.1 software (Thermo Scientific) was used to analyze mass spectra. The parameters of MS acquisition were as follows: positive mode, spray voltage 3.5 kV, S-lens RF level 50%, and capillary temperature 300 °C. The MS acquisition was performed in 200–600 Da full scan mode with a mass resolution of 70,000 [3,48]. Target MS/MS further confirmed all the possible degradation byproducts and set the collision energy at 35 eV. The characteristics of the parent compound and degradation byproducts by the MnP system are listed in Table 2.

## 4. Conclusions

The optimal medium nitrogen-limited and high level of Mn^2+^ could obtain optimal MnP activity and inhibit the expression of lignin peroxidase by *Phanerochaete chrysosporium*. The purified MnP could transform 80% tetracycline in 3 h, and the threshold of activated hydrogen peroxide was about 0.045 mmol L^−1^ in the reaction system. After the 3rd cyclic run, the transformation rate was almost the same at the low initial concentrations of TC (77.05–88.47%), while it decreased when the initial concentration was higher (49.36–60.00%). The transformation kinetics of tetracycline by MnP can satisfactorily be described by the Michaelis–Menten model. The antimicrobial potency of the transformation products by the MnP system at different initial concentrations of TC decreased throughout the reaction time. We identified seven possible degradation products and then proposed a potential TC transformation pathway, including demethylation, oxidation of the dimethyl amino, decarbonylation, hydroxylation, and oxidative dehydrogenation.

## Figures and Tables

**Figure 1 molecules-26-06803-f001:**
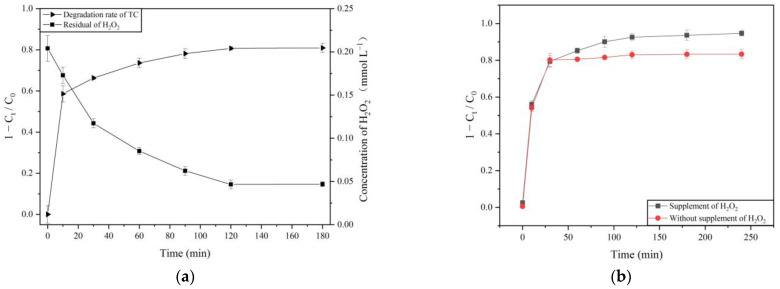
Degradation of tetracycline, residual of H_2_O_2_ (**a**), and impact of H_2_O_2_ supplementation (**b**) in MnP system.

**Figure 2 molecules-26-06803-f002:**
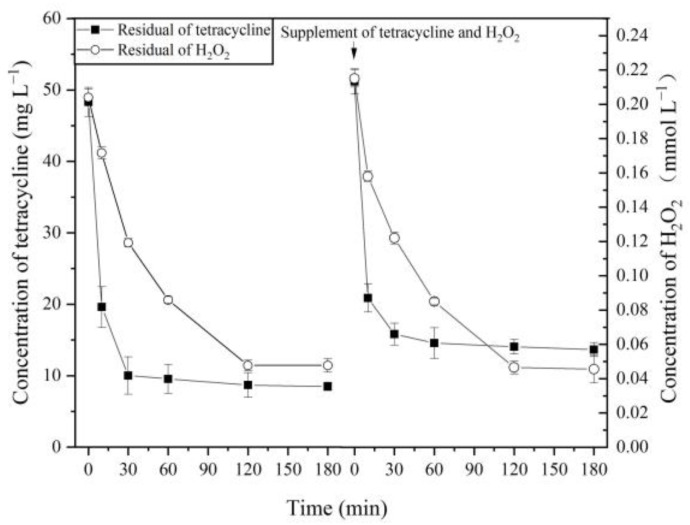
Effect of H_2_O_2_ and TC supplementation on degradation of TC by the MnP system.

**Figure 3 molecules-26-06803-f003:**
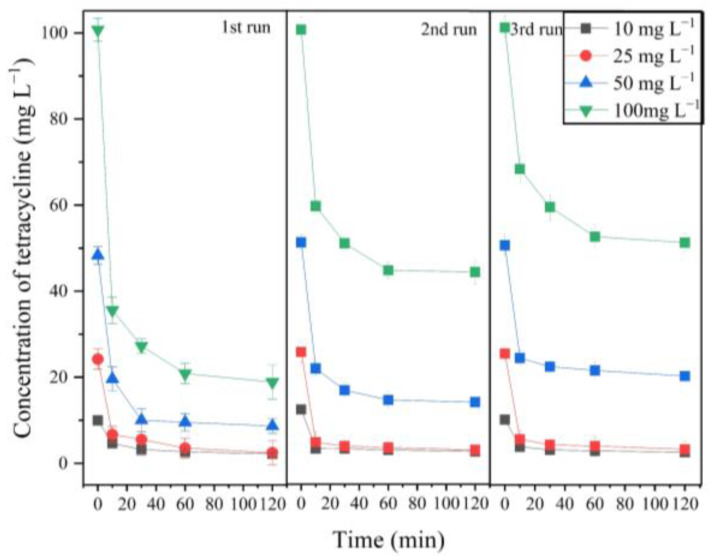
Effect of H_2_O_2_ and TC supplementation on degradation of TC by the MnP system.

**Figure 4 molecules-26-06803-f004:**
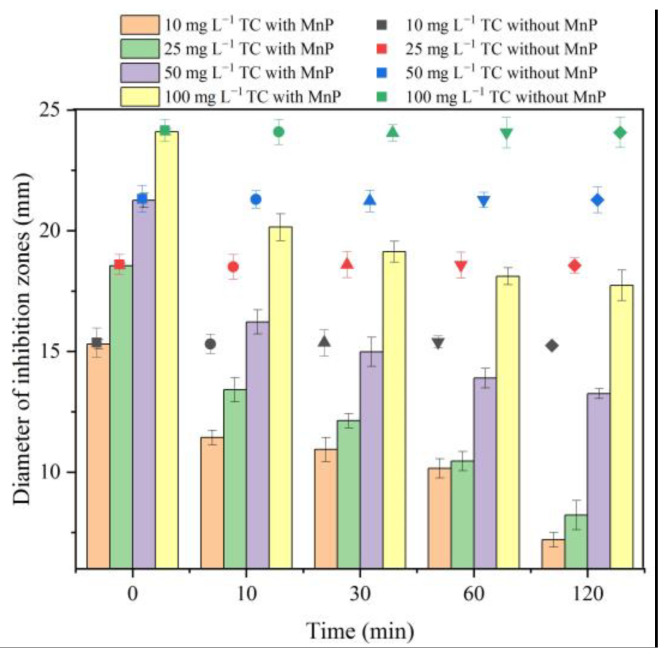
The antibacterial potency of the transformation products in MnP system were measured using inhibition zones. The inner diameter of the wells was 6 mm in the disk diffusion test.

**Figure 5 molecules-26-06803-f005:**
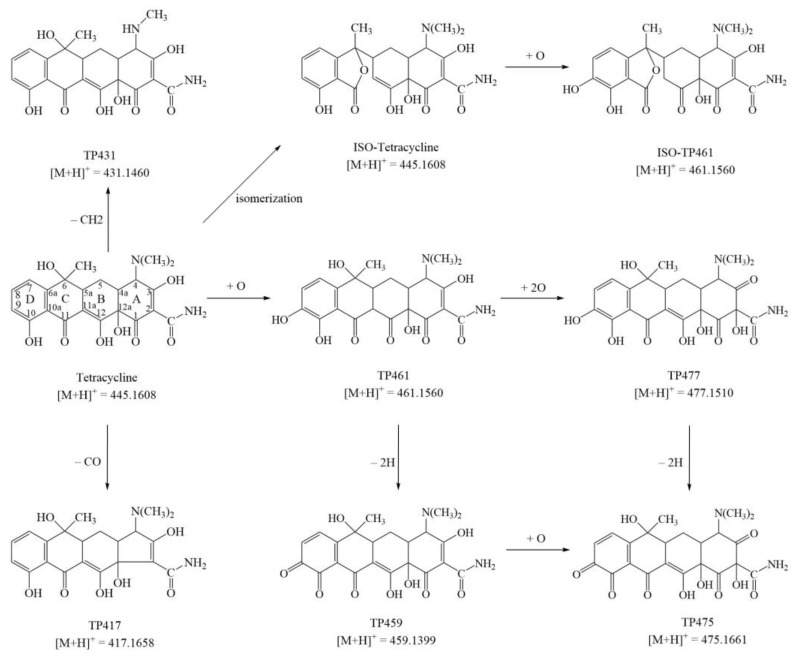
Proposed degradation pathway of tetracycline by the MnP system.

**Table 1 molecules-26-06803-t001:** Order of significance for each factor by ANOVA.

Source	DF	Adj SS	Adj MS	F-Value	*p*-Value
Glucose	1	24,200	24,200	2.78	0.171
Ammonium tartrate	1	163,020	163,020	18.74	0.012
MnSO_4_	1	12,800	12,800	1.47	0.292
Error	4	34,791	8698		
Total	1	24,200	24,200	2.78	0.171
	R^2^ = 85.18%				

DF: degrees of freedom; Adj SS: adjusted sum of squares; Adj MS: adjusted mean squares.

**Table 2 molecules-26-06803-t002:** Characteristics of the parent compound and the degradation products from the MnP system.

	Retention Time (min)	Compound	Ion	Predicted Mass (*m*/*z*)	Measured Mass (*m*/*z*)	Elemental Composition	Ring Double Bond Equivalent Value (RDB)	Intensity
Parents compound	4.19	ISO or ETC	[M+H]^+^	445.1605	445.1602	C_22_H_25_O_8_N_2_	11.5	2.69 × 10^8^
	4.72	TC *	[M+H]^+^	445.1605	445.1602	C_22_H_25_O_8_N_2_	11.5	3.33 × 10^8^
Transformation products	4.03	TP417	[M+H−CO]^+^	417.1656	417.1661	C_21_H_25_O_7_N_2_	10.5	2.52 × 10^5^
	5.26	TP431	[M+H−CH_2_]^+^	431.1449	431.1456	C_21_H_23_O_8_N_2_	11.5	1.57 × 10^7^
	3.62	ISO-TP461	[M+H+O]^+^	461.1555	461.1585	C_22_H_25_O_9_N_2_	11.5	1.09 × 10^6^
	3.80	TP461	[M+H+O]^+^	461.1555	461.1566	C_22_H_25_O_9_N_2_	11.5	1.88 × 10^6^
	5.44	TP459	[M+H+O−2H]^+^	459.1398	459.1402	C_22_H_23_O_9_N_2_	12.5	2.06 × 10^8^
	3.18	TP477	[M+H+O+O]^+^	477.1504	477.1508	C_22_H_25_O_10_N_2_	11.5	4.13 × 10^7^
	4.40	TP475	[M+H+O+O−2H]^+^	475.1347	475.1353	C_22_H_23_O_10_N_2_	12.5	3.40 × 10^5^

* TC = tetracycline.

## Data Availability

Not applicable.

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
