# Peer review of "Transformation of Tetracycline by Manganese Peroxidase from Phanerochaete chrysosporium"

_molecules, 2021, doi:10.3390/molecules26226803_

Round 1

Reviewer 1 Report

The authors have explored the potentialities of the purified MnP enzyme to degrade the tetracycline (TC) as a detoxification method in residual wastes. They determined the best culture conditions to produce the MnP enzyme as well as for the TC degradation and also proposed a potential TC transformation pathway. Also, it was demonstrated the loss of antimicrobial activity when the MnP enzyme is applied. Indeed, the work deserves publication after these minor suggestions:

  1. The influence of both glucose and MnSO4 on MnP production was not discussed
  2. Please clarify the terms DF, Adj SS and Adj MS in Table 1. Tables should be self-explanatory. I think table legends may be improved or includes foot table notes.
  3. The proposed title “Transformation mechanism of tetracycline by manganese peroxidase from Phanerochaete chrysosporium” is too ambitious for the attained results.

Author Response

Reviewer #1: The authors have explored the potentialities of the purified MnP enzyme to degrade the tetracycline (TC) as a detoxification method in residual wastes. They determined the best culture conditions to produce the MnP enzyme as well as for the TC degradation and also proposed a potential TC transformation pathway. Also, it was demonstrated the loss of antimicrobial activity when the MnP enzyme is applied. Indeed, the work deserves publication after these minor suggestions: 1. The influence of both glucose and MnSO4 on MnP production was not discussed Response: Thank you for your comments. We have discussed the influence of MnSO4 (Mn2+) on MnP production. Please see the line 181 to 189 in the manuscript. We have added the discussion about the influence of glucose on MnP production. “Glucose had no significant effect on MnP production. P. chrysosporium had the highest MnP activity under conditions of low nitrogen – high carbon, which was consistent with Trametes species [28]. This indicated that the ratio of carbon/nitrogen may synergistically affect the expression of MnP.” 2. Please clarify the terms DF, Adj SS and Adj MS in Table 1. Tables should be self-explanatory. I think table legends may be improved or includes foot table notes. Response: Thank you for your comments. The DF, Adj SS and Adj MS represent degrees of freedom, adjusted sum of squares and adjusted mean squares, respectively. We have added them as foot table notes. 3. The proposed title “Transformation mechanism of tetracycline by manganese peroxidase from Phanerochaete chrysosporium” is too ambitious for the attained results. Response: Thank you for your comments. We have appropriately converged the title. “Transformation of tetracycline by manganese peroxidase from Phanerochaete chrysosporium”

Reviewer 2 Report

I read the work presented to me for review with great interest.The work concerns a very important problem concerning the excessive use of antibiotics and, consequently, their pollution of the natural environment. In the introduction, the authors should also pay attention to the problem of resistance that pathogens acquire precisely due to the fact that antibiotics enter the environment. The work deserves to be published after a few corrections.

Line 16  “However, little is known about the transformation mechanism of manganese peroxidase (MnP) from microorganisms”. ---transformation of what? (subject is missing)

Please define TC at the beginning, because initially you have to guess that it is an antibiotic. The description of the abbreviation appears only in the table line 271.

Warto wspomnieć ze bakterie się uodparniają

Line 61 is “of vitro” should be in vitro

Line 112 and 114, 130 is  1ml  should be  1mL

Line 123 space 3min

Line 163 LiP not Lip

Line 178 is “Lip by P. Chrysosporium” should be . LiP by P. chrysosporium (italic)

Line 253 “Hence, enzyme-catalyzed antibiotics may be more suitable than physicochemical methods because it eliminates the biological activity of antibiotics.???? Please explain this, I don't really understand what the authors mean

Line 280 “N-demethyl”??

Via should be in italic

Line 283 “Hydrogen peroxide” hydrogen peroxide

Line 294 “are usually be oxidized” are usually oxidized

Line 306 Fe2O3/CeO2/ ----- Fe2O3/CeO2

Line 308 “ortho-hydroquinone” italic ortho

Line 365 Wushan ???

Author Response

Reviewer #2:

 I read the work presented to me for review with great interest. The work concerns a very important problem concerning the excessive use of antibiotics and, consequently, their pollution of the natural environment. In the introduction, the authors should also pay attention to the problem of resistance that pathogens acquire precisely due to the fact that antibiotics enter the environment. The work deserves to be published after a few corrections.

  1. In the introduction, the authors should also pay attention to the problem of resistance that pathogens acquire precisely due to the fact that antibiotics enter the environment.

Response: Thank you for your comments. The problem that antibiotic residues in the environment promote the dissemination of antibiotic resistance genes in pathogens is indeed worthy of attention. We have modified in corresponding position.

“Tetracyclines have detrimental effects on ecosystems, such as bioaccumulation along the food chain, toxicity to the microbial community, promotion the generation and dissemination of the antibiotic resistance (especially in pathogens) [10], threaten to drink source and irrigation water, and disruption of the human intestinal flora.”

  1. Line 16 “However, little is known about the transformation mechanism of manganese peroxidase (MnP) from microorganisms”. ---transformation of what? (subject is missing)

Response: Thank you for your comments. We have added subject and adjusted the sentence to make the meaning clearer.

“However, little is known about the transformation mechanism of antibiotic by manganese peroxidase (MnP) from microorganisms.”

  1. Please define TC at the beginning, because initially you have to guess that it is an antibiotic. The description of the abbreviation appears only in the table line 271.

Response: Thank you for your comments. We have defined TC at the beginning (Abstract and Introduction).

“This work investigated the transformation characteristics, the antibacterial activity of byproducts, and the degradation mechanism of tetracycline (TC) by purified MnP from Phanerochaete chrysosporium.”

“Tetracycline (TC) antibiotics are widely used in human health and veterinary medicine because of their low price and solid antibacterial ability [1].”

  1. Line 61 is “of vitro” should be in vitro

Response: Thank you for your comments. We have changed “of vitro” to in vitro.

“only a few reports demonstrated its capability of in vitro degradation of tetracycline”

  1. Line 112 and 114, 130 is 1ml should be 1mL

Response: Thank you for your comments. We have changed all the “ml” to “mL” in the manuscript.

  1. Line 123 space 3min

Response: Thank you for your comments. We have added a space between “3” and “min”.

“and recorded the absorbance increase at 238 nm within 3 min.”

  1. Line 163 LiP not Lip

Response: Thank you for your comments. We have changed “Lip” to “LiP”.

  1. Line 178 is “Lip by P. Chrysosporium” should be . LiP by P. chrysosporium (italic)

Response: Thank you for your comments. We have changed all the “Lip” to “LiP” (line 166, 180, 181 and 182).

  1. Line 253 “Hence, enzyme-catalyzed antibiotics may be more suitable than physicochemical methods because it eliminates the biological activity of antibiotics.???? Please explain this, I don't really understand what the authors mean

  Response: Thank you for your comments. Some physicochemical methods (especially photocatalysis) for the degradation of tetracycline may produce more toxic intermediates. While, the antibacterial activity of the bio-transformation intermediates of tetracycline decreased. Therefore, the biodegradation may have certain advantages from the perspective of product toxicity.

“Hence, enzyme-catalyzed antibiotics may be more suitable than physicochemical methods because enzyme can reduce the antibacterial activity of antibiotics.”

  1. Line 280 “N-demethyl”??

Response: Thank you for your comments. The word was not spelled completely. We have changed the “N-demethyl” to “N-demethylation”.

  1. Via should be in italic Line 283 “Hydrogen peroxide” hydrogen peroxide

Line 294 “are usually be oxidized” are usually oxidized

Line 306 Fe2O3/CeO2/ ----- Fe2O3/CeO2

Line 308 “ortho-hydroquinone” italic ortho

Line 365 Wushan ???

Response: Thank you for your comments. We have corrected follow the suggestions. Please track the manuscript. The “Wushan” in line 365 is a road name.